# Climate Change Adaptation in Natural World Heritage Sites: A Triage Approach

**Jim Perry**

Department of Fisheries, Wildlife and Conservation Biology, University of Minnesota, 2003 Upper Buford Circle, St Paul, MN 55108, USA; Jperry@umn.edu

**Abstract:** Climate change is a certainty, but the degree and rate of change, as well as impacts of those changes are highly site-specific. Natural World Heritage sites represent a treasure to be managed and sustained for all humankind. Each World Heritage site is so designated on the basis of one or more Outstanding Universal Values. Because climate change impacts are site-specific, adaptation to sustain Universal Values also must be specific. As such, climate change adaptation is a wicked problem, with no clear action strategies available. Further, adaptation resources are limited at every site. Each site management team must decide which adaptations are appropriate investments. A triage approach guides that evaluation. Some impacts will be so large and/or uncertain that the highest probability of adaptation success comes from a series of uncertain actions that reduce investment risk. Others will be small, certain, comfortable and yet have low probable impact on the Universal Value. A triage approach guides the management team toward highest probable return on investment, involving stakeholders from the surrounding landscape, advancing engagement and communication, and increasing transparency and accountability.

**Keywords:** risk-based decisions; triage; protected areas; scenario planning

## 1. Introduction

Human society is now living in the Anthropocene [1–5], encountering far-reaching anthropogenic environmental changes. Climate change has become one of the few most critical challenges we face [6,7]. Accelerated, anthropogenic climate change threatens the capacity of the Earth's ecosystem to sustain human well-being. The global ecosystem has a limited capacity to meet the demands of a constantly increasing human population [8,9]. As such, climate change has become the quintessential crisis, attracting public and political attention [9]. Such attention varies among fear, denial, and mitigation/adaptation.

Although globally, climate change is a certainty [10,11], the rate and magnitude of specific changes are highly site-specific. Each management entity (i.e., those responsible for management of a given landscape) must develop and implement adaptation actions specific for the site itself. That specificity is demanding, expensive and yet critical when the site involved is unique and of global significance. Natural World Heritage (NWH) sites represent a pool of 252 landscape units, each of which is unique and of global significance (Figure 1). Sixteen of those 252 sites are classified as "In-Danger" by UNESCO, and two of the sixteen (i.e., Everglades and East Rennell) are specifically threatened by climate change. Others such as the Australian Wet Tropics are highly sensitive to small and almost certain changes in climate [12–14]. Climate change is a potential threat to each NWH [15], but especially so to ecosystems that are rich in biodiversity and endemism (e.g., the subtropical rainforests of central eastern Australia) [16]. More than a decade ago, the World Heritage Committee (WHC) instructed that all World Heritage (WH) site management plans assess the possible impact of climate change and

prepare mitigation strategies [16–18]. That call has been repeated by several global groups, e.g., [19–21]. Meeting that need demands attention and resources that often are in limited supply.

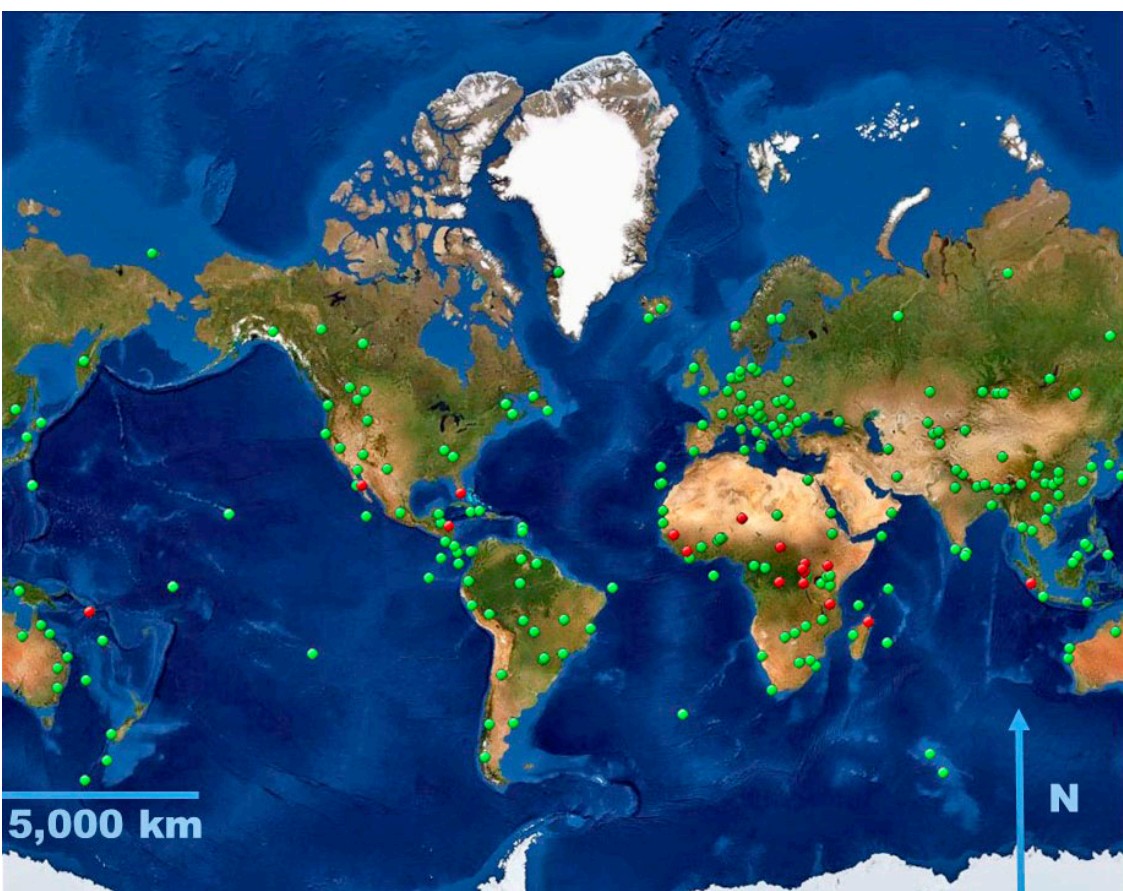

**Figure 1.** Location of the 252 UNESCO Natural World Heritage (NWH) sites; the 16 sites in red are on UNESCO's "Sites In-Danger" list [22].

Climate change, and climate change adaptation make management of NWH sites and other large protected areas ever more challenging. Sustaining the attributes that make the site qualify as World Heritage becomes pursuit of a moving target. Past and current management strategies may not be adequate for WH sites and other large protected areas in adapting to a changing climate because the sites originally were developed and managed with the notion of static boundaries and with the aim of maintaining values present at the time of designation [23]. For example, Zaccarelli et al. [24] comment that conservation plans assume that biodiversity and human values both are static. Yet, increased awareness of the value of biodiversity has changed the priorities associated with it [25]. Similarly, Wu et al. [26] demonstrate the need for forward-looking conservation strategies that can adapt to the novel communities expected to represent future avifauna, a view clearly applicable to other taxa as well.

One of the attributes of climate change that is most widely accepted is that the frequency and magnitude of extreme events will increase before there is a demonstrable change in average conditions [27]. Geologic history shows that the Earth's climate has always been characterized by high variability, underlain by natural cycles. However, as humans increasingly have altered atmospheric gases, both the frequency and intensity of events have become more pronounced, with extremes of both temperature and rainfall commonplace [11,28,29]. For example, distinct Australian flood events with 1 in 100, 1 in 150 and 1 in 300 year return periods all were recorded in 2007 [11]. That change in

climatic conditions necessitates a change in planning and management. But, because futures are so unpredictable the requirement is for an adaptable strategy, not a series of new practices.

## 2. Risks that Climate Change Poses to Natural World Heritage Sites

Each WH site has a series of attributes the cause it to meet the UNESCO criteria for WH. Those attributes represent the Outstanding Universal Value (OUV) that characterizes the site. Climate change may pose a wide range of stressors to the variables that constitute the OUV. Examples of such stressors include sea level rise, changes in temperature and precipitation, and altered hydrologic regimes. Cloud forests such as those in the mountainous regions of Western Australia and the Andes are especially sensitive to such changes [12–14,30,31]. Coral reefs such as those in Belize [32], Australia [33] and Indonesia [34] have been demonstrated to be highly responsive to climate changes. I developed a World Heritage Vulnerability Index (WHVI) for NWH sites, to identify the sites at greatest risk from climate change [35]. That index relied on nine variables, ranked all NWH sites (220 at the time), and could inform debate concerning the future management of key heritage assets [11]. However, that approach served as a comparison among sites, not as guidance for an individual site. In a second contribution [36], a colleague and I offered a range of site-specific, climate change adaptation actions within and around any given NWH site. Decisions and actions within and among NWH sites were placed in a theoretical context in a later paper [37]. The current contribution takes that work forward to offer decision-making guidance applicable to all NWH sites.

## 3. Triage as Guidance for Prioritization

Triage is a term that has been used in the medical community since the early 1900s to sort patients into categories: (1) deceased or untreatable; (2) critical; (3) stable; and (4) minor. This paper argues that decisions at the site-scale necessarily follow the logic of triage. That process for prioritizing investment of scarce resources is implicitly applied on a daily basis by managers and policymakers, but its inherent logic (i.e., specific triage application) is most often implicit [38]. For example, Reimann et al. [39] followed a triage approach to offer an index that allows ranking WH at risk from coastal hazards and found it successful. Krosby et al. [40] used large temperature gradients, high canopy cover, large relative width, low exposure to solar radiation, and low levels of human modification to calculate a riparian climate-corridor index that results in triage-based risk evaluations ranging in scale from local watersheds to the entire Pacific Northwestern US. But, by failing to be explicit and transparent, many decisions will necessarily be inefficient [41].

Disaster risk reduction (DRR) is a large and critically important field that attempts to guide society in preparing for and responding to natural disasters. Human society has always been subject to disaster, a history which logically would lead to advanced states of prediction and planning. In fact, however, society broadly, and many governments in particular appear to have short memories of past disasters and very low willingness to invest in disaster avoidance [42]. Science and policy have come to recognize that integrating disaster response and climate change adaptation is a necessity [43]. A wide range of frameworks has been developed for framing societal actions [44,45]. In spite of that breadth, the linkage between DRR and climate change response has yet to emerge. For example, at the time of this writing (21 August 2019), Web of Science reported 3474 papers that addressed "disaster risk reduction", but only six of those addressed climate change. Only one [46] addressed climate change and triage, and the context was medical response not preparedness or adaptation. In spite of the logical similarity between DRR preparedness and climate adaptation planning, the frameworks advanced for the former are not useful for site-specific or landscape-scale climate change adaptation planning for NWH.

Bottrill et al. [41] suggest that conservation triage can serve effectively for prioritizing actions as long as alternative actions are assessed relative to at least four parameters: values, biodiversity benefit, probability of success, and cost, necessarily combining those four in a mathematically rigorous fashion. Of course, understanding the probability of success requires a clear overview of the actual

situation and clear future risk scenarios. Further, triage and transparency by themselves will not solve the problem. I have suggested that climate change adaptation at NWH sites is a wicked problem for a variety of reasons [37], including the fact that there is an ever-changing baseline. Wicked problems have a range of characteristics that challenge climate change decision-making: each manager is faced with limited resources, competing public interests, increasing and novel threats, changing political environments and demands from a diversity of stakeholders [23]. Many of those novel threats will be unknown (in frequency and magnitude), making it difficult to judge probable successes. In every NWH site, some climate change impacts will be unmanageable given available resources, and that will prevent site managers from meeting the goal of sustaining all values [23].

Philips [16] suggests that one of the principal challenges of adaptation is that there are no limits to the time and resources that could be absorbed and the absence of a limit makes it difficult to know how much to spend on adaptation. Further, there are widespread concerns about data reliability and the contested nature of information about climate change, resulting in a perceived lack of the necessary information to make fully informed decisions [16].

It is becoming increasingly clear that NWH sites and protected areas (PAs) are social-ecological systems [47]. As such, these resources are vulnerable to political change [48–50], economic fluctuations, and ecological variance [47] (Figure 2). Protected areas (including NWH sites) cannot be effectively managed based solely on ecological principles and the surrounding environment in which they are embedded [50]. Biophysical processes (e.g., landscape evolution, climate change) as well as sociopolitical dynamics drive questions like what specifically should be preserved, why and how? The most specific and practical guidance available for site-specific climate adaptation at individual NWH sites is the UNESCO Guide [36]. That guide and the paper offered here frame decisions as risk. Explicit triage logic guides decisions about such risks.

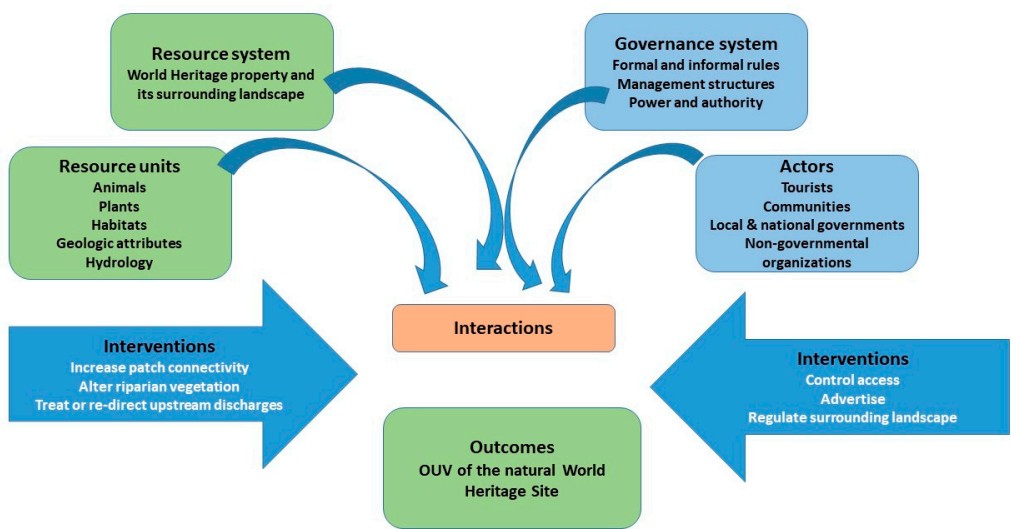

**Figure 2.** Biophysical and socioeconomic influences on the Outstanding Universal Value (OUV) of a NWH site, and examples of interventions as part of climate change adaptation. Modified from [47].

Every WH site has one or more OUVs. An OUV implies cultural and/or natural significance which is so exceptional that it transcends national boundaries and is " . . . cultural and/or natural significance which is so exceptional as to transcend national boundaries and to be of common importance for present and future generations of all humanity." [22]. The reason for management to consider climate change adaptation is to ensure that the OUV of the WH site is sustained, which requires resilience in the face of climate change. That requires that we address at least two key biophysical factors for assessing the impact of climate change: (1) the character of the site within the natural landscape; and (2) the sensitivity and vulnerability of that natural landscape to process-driven, geomorphologic change [9]. But management decisions must consider more than that. Every decision must weigh the

degree to which maintenance and intervention can be deemed cost-effective (based on the state party's definition) and the degree to which adaptation actions will be compatible with the aesthetic and other qualities that are part of the OUV [11].

It is appropriate that climate change adaptation at NWH sites focuses on the OUV. However, that focus cannot be blind to the natural processes of landscape change. Kabat et al. [51] offer three broad challenges that should guide such awareness: (1) conservation in a changing system requires a focus on the values and not the state of the system; (2) local and regional management should be nested in more coarse-scale governance structures; and (3) the management approach must be sufficiently adaptable that it is able to deal with uncertainty and nonlinearities.

## 4. Guidance for Triage-Based Decisions

A site manager, or the site management team must be prepared to take calculated risks and choose optimal solutions in the context of what is known and understood. A triage approach supports that. Triage guides allocation of conservation resources such that scarce resources are allocated to maximize persistence of features that will disappear without conservation action [37]. That is, based on climate predictions and scenarios of the future, some attributes of the site (and of the OUV) will be relatively unaffected, some will almost surely be altered in spite of management action, and some are likely to be responsive to (protected by) management action. Investing time and resources in the latter of those three maximizes the probability of a successful outcome (Figure 3). Assessing the vulnerability of a site's features against climate scenarios will help the site team assess the degree of risk climate change poses to the site's OUV. Scenarios incorporate interacting risks and uncertainties, advancing informed decisions [52]. This in turn, will enable the management team to prioritize responses against criteria, providing a basis for action that can be monitored and reviewed. By explicitly acknowledging that scenarios and decisions are based in triage, the manager will be able to evaluate tradeoffs and provide transparency of decision-making [39].

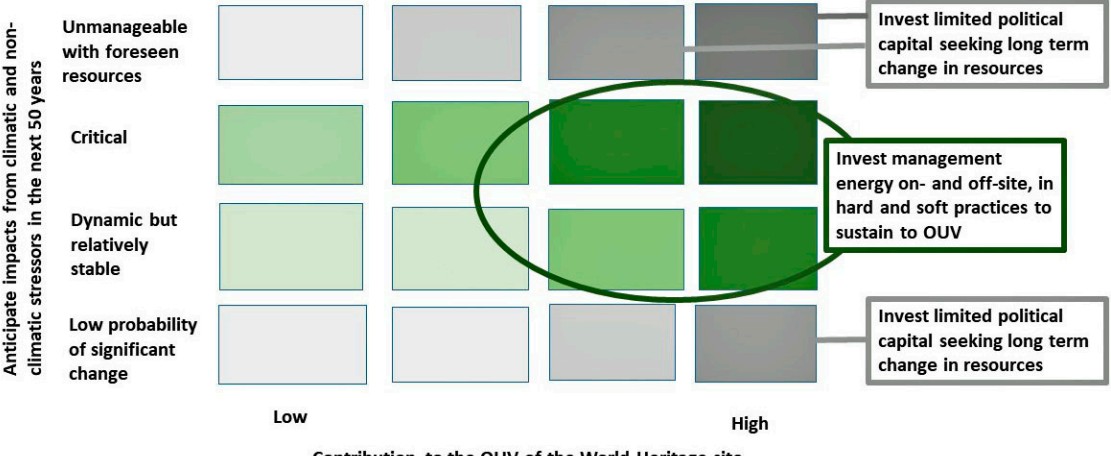

**Figure 3.** A risk triage for managing the OUV of a NWH site. Modified from [38]. For example, a strategy in the lower right (low risk, high value) might be to expand the boundaries of the site, a low probability effort that would require significant time and energy. Strategies in the upper right represent high value attributes that probably cannot be saved (e.g., eroding coastlines). This approach recognizes that valuable resources (i.e., upper right corner) will be lost without additional resources, but argues that long term sustainability of the OUV comes from investment in the right center. Examples of such investments might include wildlife corridors to connect patches, or no-take marine protected zones for reproduction.

While future climate conditions are very difficult to predict precisely, even rough predictions in the form of scenarios will help a manager think about the ways in which attributes of the OUV may be

expected to respond to future climate conditions. This allows at least some form of risk analysis as the basis for designing an adaptation plan. Such a plan should provide a range of prioritized actions, both within and beyond the site itself. Those actions should include actions to be taken or proposed in the broader landscape. The most successful climate change adaptation strategies view the site as an element of a larger landscape and then address the OUV on-site in the context of off-site practices that influence the OUV [36].

Examples of such on- and off-site impacts abound. The Sundarbans is a massive wetland complex in India and Bangladesh. Some off-site impacts (e.g., sea level rise) are beyond the consideration of the management team. Others (e.g., deforestation in the headwaters, mining in adjacent properties) can be influenced by but not controlled by the management team. On-site practices such as hunting and shrimp culture can be directly controlled; others, such as on-site effects of flooding cannot [53]. However, traditional management has viewed the OUV of the site as the operational variable and the site boundaries as the spatial sphere of interest. A triage approach that considers off-site and on-site factors, and uses risk analysis logic to guide investments might, for example, guide the team toward mining and shrimp culture as the two most opportune investments, and expansion of site boundaries as one viable, long term strategy for moving forward.

The team should recognize that climate (and other anthropogenic) changes in the larger landscape will make future protected area management even more challenging than traditional management. Many traditional techniques will not be adequate for future conditions because NWH sites have been developed and managed in conceptual isolation, with the notion of static boundaries and the aim of maintaining current values [23,37]. It is common for managers and their teams to focus on lands under their control (or influence). For example, Fischman et al. [54] found that few protected area managers prescribed acting outside the refuge to address climate change impacts. They report that their finding is particularly surprising because protected area managers showed overall high rates of prescriptions for acting outside of refuge boundaries to address other problems, especially addressing water pollution, habitat loss, and invasive species. Yet protected areas (including NWH sites) are part of regional zones of social-ecological interactions. Management actions inside a site (e.g., ecological connectivity, water flows) impact, and are impacted by responses (e.g., development) outside site boundaries [55]. Actions outside site boundaries are often controlled by political, bureaucratic, administrative or ownership variables, but cannot be ignored by site managers. As Kabat et al. [51] and Jones et al. [56] suggest, global trends in the economy cause an increasing need to evaluate projects on more coarse spatial scales; I would add the need for expanded temporal scales.

Specifically with regard to temporal influences, the site team must think into the future. Managers are less able than ever to predict what will happen under future climate conditions, but a diverse team can frame scenarios and make educated guesses. Climate change adaptation requires analysis of the current situation in light of historical and projected changes, measuring the results of actions taken, revising them and trying again. Adaptive management is based on this cycle of analysis, application, evaluation and revision [41]. That forward looking approach may seem intuitive, but is not traditional. Many of the recommendations for improving climate change adaptation encourage managing for change and adopting landscape-scale strategies [23]. Global changes such as increased mobility of many species, and increases in human alteration of the landscape mean that NWH sites can no longer be managed as ecological islands, independent of the broader social-ecological system in which they are located. Resilience of a site requires an ability to adapt to changing social and ecological conditions over time, supporting persistence of populations, communities, and ecosystems inherent to the OUV [47].

Adaptation practices at a NWH site are implemented at different levels. Site-specific, lower-level actions can be implemented by the management team responsible for the site. However, each site is nested within a larger biophysical and socioeconomic landscape that strongly influences site conditions. Higher-level actions involving stakeholders such as the surrounding community, policy-makers or energy and water companies also need to be considered. Practices at the site level are generally

less expensive and provide quicker responses but may have more limited impact on protecting the OUV [36]. Critically, however, each decision to implement a practice should be framed as a triage, recognizing that various actions have different probabilities of impact, and each action represents a tradeoff, an investment of resources that could be used elsewhere.

## 5. Sustaining an Evolving Outstanding Universal Value: Manage with Transparency and Accountability

In initiating a triage approach, the NWH site management team will initially focus on the OUV; if the protected area is not a WH site, there will be a similar logic involving identified attributes deserving protection. The OUV and its attributes provide a framework which by itself does not offer enough detail for guiding adaptive responses. Vulnerability, the probability of climate change damage to the attributes of the OUV, or to ecosystem resilience provides the risk estimate. Vulnerability assessments go beyond threat assessments in that they help the management team understand the capacity of the site to withstand or adapt to climate change (and other anthropogenic) impacts [54]. Climate change vulnerability specifically identifies the extent to which predicted climatic conditions are likely to cause a negative impact to the site's OUV. Expressions of vulnerability necessarily include cultural processes (e.g., socioeconomic influences, adaptation and mitigation actions, governance). The site team must consider those variables through engaging stakeholders in the surrounding landscape. As NWH site managers become more aware of climate impacts, the team must become more tightly connected with regional and national governance structures [51]. Inherent in that increase in scale and increase in inter-jurisdictional linkages will be the need to address and function under conditions of increased uncertainty [51]. The increasingly coarse spatial and temporal scale, the increased institutional complexity and increased uncertainty support the argument that this is a wicked problem [37] but also support the argument that a triage approach is necessary. The team should ensure that management goals are well communicated with all relevant stakeholders including those in the WH system (e.g., UNESCO, national government) and principal parties in the regional landscape. That transparency results in broader buy-in and wider acceptance of and support for decision-making under uncertainty. Decisions taken and actions performed should be monitored and evaluated such that the broader community interprets team behavior with a sense of accountability. No management team will be able to meet all expectations. Encouraging transparency and accountability provides the support necessary to take risky decisions.

## 6. Conclusions

A site manager, or the site management team for a NWH site must be prepared to take calculated risks and choose optimal solutions in the context of what is known and predicted about socioeconomic and biophysical conditions and future climates. Those decisions will be made under high uncertainty. Successful and sustainable management of a NWH will include both the site and the surrounding landscape, including the communities in that landscape. Decision-making under uncertainty will be most successful if the stakeholders have a strong sense that management is transparent and operates with a sense of accountability. Triage decision-making helps managers frame decisions in an explicit context that accepts the uncertainty involved, and advances both transparency and accountability.

**Funding:** This research was funded by the University of Minnesota.

**Acknowledgments:** I am grateful to Chiara Bertolin and three anonymous reviewers for helpful comments on earlier drafts of this manuscript.

**Conflicts of Interest:** The author declares no conflict of interest.

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
