# Peer review of "Climate Change Adaptation in Natural World Heritage Sites: A Triage Approach"

_climate, doi:10.3390/cli7090105_

Round 1
Reviewer 1 Report
Comment: Found this a very interesting article and very relevant, with the triage argument and decision making in regards to WH management needing this kind of attention, very well written.
Recomendations
Line 45 - Believe a better example and more appropriate reference, in Australia WH Wet Tropics is highly vulnerable with many studies - only 1 degree may result in significant changes to forest ecosystems and species and high extinction rate (Hilbert et al. 2001; Ostendorf et al. 2001; Krockenberger et al. 2003; Williams et al. 2003; Meynecke 2004)
Felt some of the WH sites most at risk are cloud forest which seems to be missing from the article
Author Response
Reviewer 1
Comment: Found this a very interesting article and very relevant, with the triage argument and decision making in regards to WH management needing this kind of attention, very well written.
Recommendations
Line 45 - Believe a better example and more appropriate reference, in Australia WH Wet Tropics is highly vulnerable with many studies - only 1 degree may result in significant changes to forest ecosystems and species and high extinction rate (Hilbert et al. 2001; Ostendorf et al. 2001; Krockenberger et al. 2003; Williams et al. 2003; Meynecke 2004)
This information, and the relevant citations have been addedFelt some of the WH sites most at risk are cloud forest which seems to be missing from the article
This information, and the relevant citations have been added
Reviewer 2 Report
The author has made an effort to propose a matrix based on triage approach for prioritizing investment for climate change adaptation in NWH. Here are my comments.
I personally, do not like acronyms in abstract. Please do not use same keywords as in title. This limits searchability of your article, as same are already present in title. L29-30. Rephrase. WH? Mention abbreviation beforehand. Can author add more about international agreements, conventions or agendas highlighting need for mitigating climate change for natural world heritage conservation? Please re-write as “This study proposes a ….” I would ask author to refrain using this study proposes these indicators/index, as proposed the indicators/index is already published, as referred by author. This makes author’s stance of proposition of index incorrect. Figure 1. How many are classified as in “danger” or “high risk” by UNESCO? Are they threatened by CC? How does author’s classification of “at risk” sites match with UNESCO’s “danger” sites? In relation to previous comments, Table 1 seems unnecessary. Does it reinforce author’s triage approach? How does the author classify climate change impacts on NWH, into 1) untreatable 2) critical, 3) stable and 4) minor? Can author add give some critical insight into potential “climatic stressors”. The matrix looks vague, unless author provides which impact can be classified as unmanageable or stable”. A description of four classes needs to be explicitly discussed, as well as the three investment strategies in figure 3. Based on complexity of natural landscape and biodiversity, these stressors will be different for each type of NWH. Then how can these be relatively gauged? I see major flaws in operationalizing of the proposed approach. Avoid abbreviations in headings. Numerous frameworks are constructed in disaster risk reduction literature, for reducing risks to cultural heritage sites. How does against authors stance on “climatic stressors” relates with DRR frameworks. Can author generate a critical debate on climate change adaptation investment against disaster risk reduction investment?
References
Taboroff, J. (2000). Cultural heritage and natural disasters: incentives for risk management and mitigation. Managing Disaster Risk in Emerging Economies. New York: World Bank. Disaster Management Risk, 2, 71-79. O'Brien, G., O'Keefe, P., Jayawickrama, J., & Jigyasu, R. (2015). Developing a model for building resilience to climate risks for cultural heritage. Journal of Cultural Heritage Management and Sustainable Development, 5(2), 99-114. Burns, W. C. (2009). Belt and Suspenders? the world heritage convention's role in confronting climate change. Review of European Community & International Environmental Law, 18(2), 148-163. Harrison, R. (2015). Beyond “natural” and “cultural” heritage: toward an ontological politics of heritage in the age of Anthropocene. Heritage & Society, 8(1), 24-42. Wei, J., Zhao, Y., Xu, H., & Yu, H. (2007). A framework for selecting indicators to assess the sustainable development of the natural heritage site. Journal of Mountain Science, 4(4), 321-330.
Author Response
Reviewer 2
I personally, do not like acronyms in abstract.
This has been correctedPlease do not use same keywords as in title. This limits searchability of your article, as same are already present in title.
This has been correctedL29-30. Rephrase. WH? Mention abbreviation beforehand.
This has been correctedCan author add more about international agreements, conventions or agendas highlighting need for mitigating climate change for natural world heritage conservation?
Three additional sources have been addedPlease re-write as “This study proposes a ….” I would ask author to refrain using this study proposes these indicators/index, as proposed the indicators/index is already published, as referred by author. This makes author’s stance of proposition of index incorrect.
This has been correctedFigure 1. How many are classified as in “danger” or “high risk” by UNESCO? Are they threatened by CC? How does author’s classification of “at risk” sites match with UNESCO’s “danger” sites?
Sixteen natural sites are in danger, and two of those are specifically threatened by climate change. The text has been addedIn relation to previous comments, Table 1 seems unnecessary. Does it reinforce author’s triage approach?
Table 1 was added at the request of an external reviewer. It does not specifically relate to the triage approach and has been removed.How does the author classify climate change impacts on NWH, into 1) untreatable 2) critical, 3) stable and 4) minor?
This classification is not proposed for or appropriate for use at the scale of an entire WH site. It is applied to specific variables within a World Heritage landscape. The management team has the responsibility to ask how climate change will affect on-site variables, and which of those effects fall into each of the four categories.Can author add give some critical insight into potential “climatic stressors”.
Insights and examples have been addedThe matrix looks vague, unless author provides which impact can be classified as unmanageable or stable”. A description of four classes needs to be explicitly discussed, as well as the three investment strategies in figure 3.
The legend for the figure has been expanded to include the requested detail. There are not four discrete classes, but rather an illustrative gradation from low to high value. The examples in the legend now make that more clear.Based on complexity of natural landscape and biodiversity, these stressors will be different for each type of NWH. Then how can these be relatively gauged?
The approach proposed her specifically is applicable at the scale of individual sites, not among sites. There is no attempt to “relatively gauge” stressors among sites but rather to gauge stressors influencing variables within a site, and guiding decisions in response.I see major flaws in operationalizing of the proposed approach.
This comment does not have sufficient detail to be addressed. The text has been corrected in many places in response to reviews, making the approach more operationalAvoid abbreviations in headings.
This has been correctedNumerous frameworks are constructed in disaster risk reduction literature, for reducing risks to cultural heritage sites. How does against authors stance on “climatic stressors” relates with DRR frameworks. Can author generate a critical debate on climate change adaptation investment against disaster risk reduction investment?
The citations the reviewer offered have been reviewed and the relationship between DRR frameworks and what is proposed here for World Heritage and climate change has been elaborated.
Reviewer 3 Report
General comments
The article „Climate change adaptation in Natural World Heritage sites: A triage approach” has an interesting topic related to NWH management. However, the writing structure of the article doesn’t look like proper (a Conclusions section is required!), in particular, the list of references which can be improved in the context of such a well-documented subject. Accordingly, the article should be revised in a proper form that contains the key results of the study as recommended as above.
Other comments are as follows:
Line 11: The abbreviations should be defined in parentheses the first time they appear in the abstract, main text and in figure or table captions and used consistently thereafter: (…) Natural World Heritage (NWH) sites represent a treasure to (…) Lines 28 – 29: This phrase must be supported by more than one bibliographic reference in the context of such a debated topic. Line 36: Text is not formatted correctly (delete the space between paragraphs – line 36). See the Manuscript Preparation and use the manuscript template. Lines 41-42: The abbreviations should be defined in parentheses the first time they appear in the abstract, main text and in figure or table captions and used consistently thereafter: (…) NWH sites represent a pool of little more than (…) Line 46: The abbreviations should be defined in parentheses the first time they appear in the abstract, main text and in figure or table captions and used consistently thereafter: (…) World Heritage Committee (WHC) instructed that all WH site (…) Figure 1: All figures must contain: legend, coordinate grid, the north arrow, scale bar, and visible text and lines. Also, the details related to references must be deleted from the figure caption and used only the reference number. See the Reference List and Citations Guide for more detailed information. Line 67: Text is not formatted correctly (delete the space between paragraphs – line 67). See the Manuscript Preparation and use the manuscript template. Line 81: The abbreviations should be defined in parentheses the first time they appear in the abstract, main text and in figure or table captions and used consistently thereafter: (…) I have proposed a World Heritage Vulnerability Index (WHVI) for natural WH (…) Lines 87 – 89: (…) natural WH site. Decisions and actions within and among natural WH sites were placed in a theoretical context in a later paper (…) Line 91 – 92: Please use the abbreviations in the table caption. (WHVI; WH) Line 96: “Triage” instead of “Triage” Line 109: Text is not formatted correctly (delete the space between paragraphs – lines 109, 126, 133, 149, 163., 173 … ). See the Manuscript Preparation and use the manuscript template. Line 116 – 117: Please use the abbreviations (WH) Line 150: Please use the abbreviations (WH) Line 154: Please use the abbreviations (WH) Figure 3 caption: Text is not formatted correctly. See the Manuscript Preparation and use the manuscript template. Line 232: Please use the abbreviations (WH) Lines 271 – 299: Text is not formatted correctly. See the Manuscript Preparation and use the manuscript template A Conclusions section is required!Author Response
Reviewer 3
The article „Climate change adaptation in Natural World Heritage sites: A triage approach” has an interesting topic related to NWH management. However, the writing structure of the article doesn’t look like proper (a Conclusions section is required!), in particular, the list of references which can be improved in the context of such a well-documented subject. Accordingly, the article should be revised in a proper form that contains the key results of the study as recommended as above.
Other comments are as follows:
Line 11: The abbreviations should be defined in parentheses the first time they appear in the abstract, main text and in figure or table captions and used consistently thereafter: (…) Natural World Heritage (NWH) sites represent a treasure to (…)
This has been correctedLines 28 – 29: This phrase must be supported by more than one bibliographic reference in the context of such a debated topic.
Additional citations have been addedLine 36: Text is not formatted correctly (delete the space between paragraphs – line 36). See the Manuscript Preparation and use the manuscript template
Entire text has been reformattedLines 41-42: The abbreviations should be defined in parentheses the first time they appear in the abstract, main text and in figure or table captions and used consistently thereafter: (…) NWH sites represent a pool of little more than (…)
This has been correctedLine 46: The abbreviations should be defined in parentheses the first time they appear in the abstract, main text and in figure or table captions and used consistently thereafter: (…) World Heritage Committee (WHC) instructed that all WH site (…)
This has been correctedFigure 1: All figures must contain: legend, coordinate grid, the north arrow, scale bar, and visible text and lines. Also, the details related to references must be deleted from the figure caption and used only the reference number. See the Reference List and Citations Guide for more detailed information.
The figure has been updated to meet formatting guidelinesLine 67: Text is not formatted correctly (delete the space between paragraphs – line 67). See the Manuscript Preparation and use the manuscript template.
This has been correctedLine 81: The abbreviations should be defined in parentheses the first time they appear in the abstract, main text and in figure or table captions and used consistently thereafter: (…) I have proposed a World Heritage Vulnerability Index (WHVI) for natural WH (…)
This has been correctedLines 87 – 89: (…) natural WH site. Decisions and actions within and among natural WH sites were placed in a theoretical context in a later paper (…)
This has been correctedLine 91 – 92: Please use the abbreviations in the table caption. (WHVI; WH)
This has been correctedLine 96: “Triage” instead of “Triage”
This has been correctedLine 109: Text is not formatted correctly (delete the space between paragraphs – lines 109, 126, 133, 149, 163, 173 … ). See the Manuscript Preparation and use the manuscript template.
This has been correctedLine 116 – 117: Please use the abbreviations (WH)
This has been correctedLine 150: Please use the abbreviations (WH)
This has been correctedLine 154: Please use the abbreviations (WH) Figure 3 caption: Text is not formatted correctly. See the Manuscript Preparation and use the manuscript template.
This has been correctedLine 232: Please use the abbreviations (WH)
This has been correctedLines 271 – 299: Text is not formatted correctly. See the Manuscript Preparation and use the manuscript template
This has been correctedA Conclusions section is required!
A conclusion section has been added
Round 2
Reviewer 3 Report
The paper has been considerably improved according to all the recommendations made and I suggest accepting the paper to be published in the present form. I still have one recommendation related to the conclusions section, but this issue should be clarified by the Editor.
Best regards!